# The emergence of macroscopic currents in photoconductive sampling of optical fields

Johannes Schötz[1,2 ✉], Ancyline Maliakkal[1,2], Johannes Blöchl [1,2], Dmitry Zimin[2], Zilong Wang[1,2], Philipp Rosenberger [1,2], Meshaal Alharbi [3], Abdallah M. Azzeer [3], Matthew Weidman[1,2], Vladislav S. Yakovlev [1,2], Boris Bergues[1,2] & Matthias F. Kling [1,2,4,5 ✉]

Photoconductive field sampling enables petahertz-domain optoelectronic applications that advance our understanding of light-matter interaction. Despite the growing importance of ultrafast photoconductive measurements, a rigorous model for connecting the microscopic electron dynamics to the macroscopic external signal is lacking. This has caused conflicting interpretations about the origin of macroscopic currents. Here, we present systematic experimental studies on the signal formation in gas-phase photoconductive sampling. Our theoretical model, based on the Ramo–Shockley-theorem, overcomes the previously introduced artificial separation into dipole and current contributions. Extensive numerical particle-in-cell-type simulations permit a quantitative comparison with experimental results and help to identify the roles of electron-neutral scattering and mean-field charge interactions. The results show that the heuristic models utilized so far are valid only in a limited range and are affected by macroscopic effects. Our approach can aid in the design of more sensitive and more efficient photoconductive devices.

[1] Department of Physics, Ludwig-Maximilians-Universität Munich, D-85748 Garching, Germany. [2] Max Planck Institute of Quantum Optics, D-85748 Garching, Germany. [3] Attosecond Science Laboratory, Physics and Astronomy Department, King Saud University, Riyadh 11451, Saudi Arabia. [4] SLAC National Laboratory, Menlo Park, CA 94025, USA. [5] Applied Physics Department, Stanford University, Stanford, CA 94305, USA. ✉email: johannes.schoetz@mpq.mpg.de; kling@stanford.edu

Intense few-cycle laser pulses can induce conductivity within a fraction of an optical cycle, enabling the ultrafast manipulation of electric currents. Such optical-field driven currents have been measured in various materials[1–5] and 2D-structures[6]. Using sub-cycle current injection as a temporal gate, in a generalization of the concept of THz photoconductive field-sampling[7,8], the field-resolved measurement of optical waveforms up to PHz-frequencies has been demonstrated[9]. The investigation and control of these field-driven currents does not only promise to revolutionize the field of attosecond physics, but potentially also impacts technology.

For the investigation of field-dependent processes, knowledge of the carrier-envelope phase (CEP) is needed for an unambiguous determination of the field. For optical frequencies, techniques such as attosecond streaking[10] and the stereo above-threshold-ionization (stereo-ATI) phase meter[11] can be used to determine the electric field waveform and CEP, respectively. They require, however, complex ultra-high vacuum setups, in which electron time-of-flight spectra can be recorded. This has been limiting their widespread application in many laboratories.

With photoconductive field sampling in solids, a much simpler solution has been presented. Recently, the same concepts have been applied in air for the measurement of the CEP[12] and electric field[13,14], offering additional advantages: they are easier to use since they eliminate the requirement of sample fabrication and inherently use a refreshable target. Moreover, the microscopic response at the atomic/molecular level can be numerically calculated using the time-dependent Schrödinger equation, and may be modeled and interpreted in the framework of the strong-field approximation (SFA) and the simpleman's model (SMM)[15]. The general experimental setup of strong-field sub-cycle controlled currents is almost identical to broadband THz generation in gases[16–18] and both processes are expected to be closely linked.

Despite the importance of a detailed understanding of the signal formation in ultrafast current measurements, the macroscopic aspects are not yet fully understood. Even though processes such as electron scattering are expected to play an important role[13,14], there is no rigorous model that connects the microscopic single-electron dynamics to the macroscopic current signal measured on the electrodes. Apart from refs. [19,20] where the role of the electrode distance was investigated, no systematic studies exist.

An overview over existing, purely heuristic, models for macroscopic signal generation in ultrafast current sampling in gases, together with the expected dependence on pressure is shown in Fig. 1. Note that all these rather simple models ignore the Coulomb interaction between light-created charges. In the first model, a few-cycle optical laser pulse can directly induce CEP-dependent strong-field photoemission from one of the electrodes, resulting in a corresponding current (cf. Fig. 1a). Here, the sign of the current would be opposite if the counter-electrode is illuminated. Generally, this contribution is expected to decrease from vacuum toward higher pressures (black line in Fig. 1d) due to scattering-limited propagation of the released electrons. In most ultrafast current sampling experiments electrode photoemission produces a background that is avoided. The second heuristic model is denoted as photocurrent and shown in Fig. 1b. Here, not the electrodes but the gas medium between them is (tunnel)ionized. The same laser field then accelerates ions and free electrons toward the electrodes. The current depends on the laser field, e.g., for a certain CEP more electrons impinge on the left detector than on the right (as shown in Fig. 1b) or vice versa when introducing a shift of $\pi$ in CEP. The blue line in Fig. 1d shows the expected pressure dependence. At low pressures, where the mean-free path is larger than the distance to the electrodes, the number of detected charges is proportional to the medium density and the

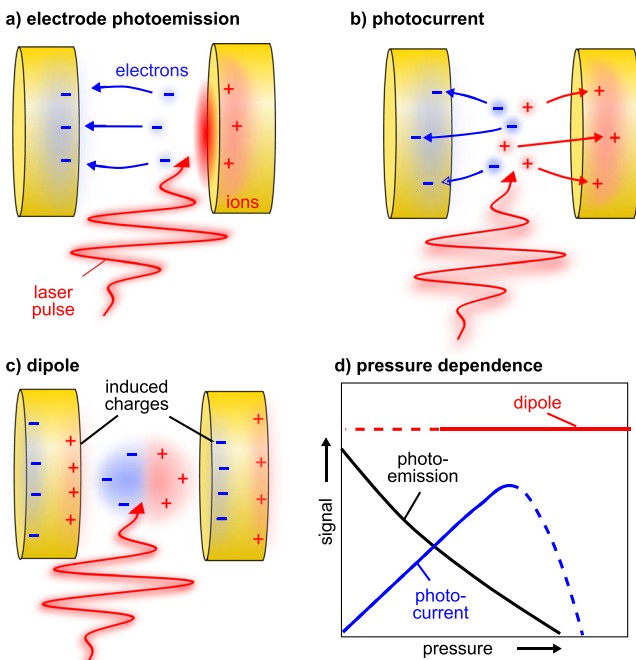

**Fig. 1 Heuristic models of signal formation in ultrafast current sampling in gases. a** Electrode photoemission, **b** photocurrent, and (**c**) dipole. The electrodes are depicted in yellow, where individually a current toward ground can be measured. **d** Expected pressure dependence for the mechanisms depicted in (**a–c**), where Coulomb interaction is neglected.

current grows linearly with pressure. For high enough pressures, charges cannot reach the detector anymore before scattering and in turn lose their strict relationship to the CEP. As a result, the macroscopic current would be expected to drop sharply. Previous work discussed whether this contribution can play a role at ambient pressure[13]. The third heuristic model involves a laser-induced charge dipole following the ionization of the medium, cf. Fig. 1c. This induces an image charge on the electrodes, yielding a macroscopic current. Interestingly, in this case, a rather constant current over pressure is expected as depicted as red line in Fig. 1d, since the total charge scales linearly with pressure, while the mean-free path scales inversely, keeping the dipole strength constant. While these heuristic models have been invoked in the interpretation of previous results[13,14,21], their validity has been debated and they failed to provide quantitative predictions. Clearly, to further advance our understanding of ultrafast photoconductive sampling in gases, a quantitative model for the macroscopic signal formation is crucial.

Here, we present a rigorous theoretical approach based on the Ramo–Shockley theorem (cf. refs. [22–25]), which avoids the artificial separation into photocurrent and dipole contributions and overcomes the limitations of the existing heuristic models. Our extensive numerical particle-in-cell (PIC)-type simulations based on this model enable a quantitative comparison with the experimental results, and enable identifying the roles of electron-neutral scattering and mean-field charge interactions, thereby providing a fundamental understanding of photoconductive sampling of optical fields. The ability of the model for quantitative predictions paves the way toward the implementation of more sensitive and more efficient petahertz optoelectronic devices.

## Results

**Experimental setup.** In our experiment (see Methods for details), laser pulses of 4.5 fs duration at 750 nm are focused between two

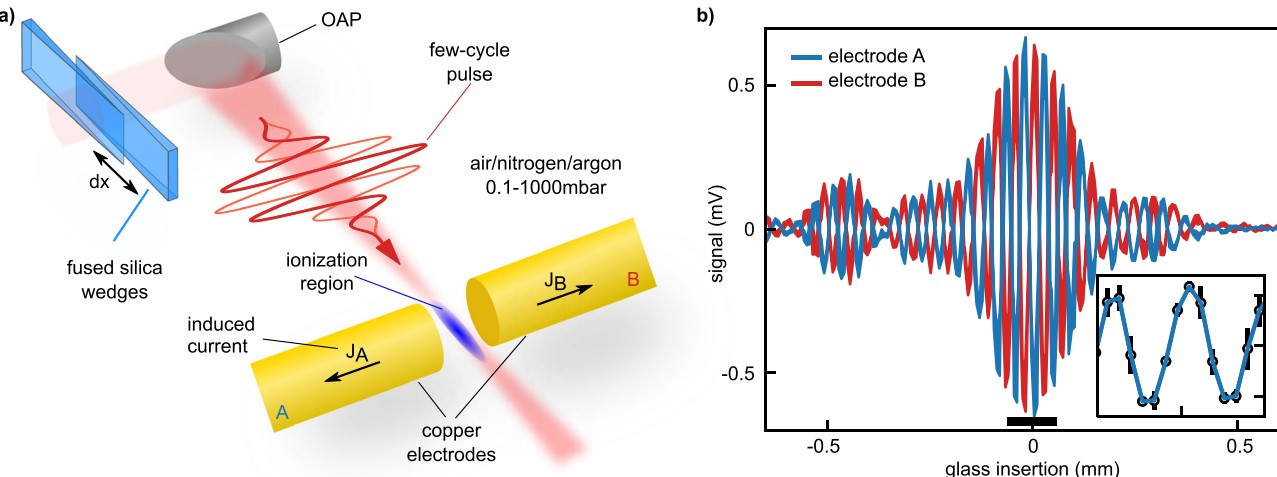

**Fig. 2 Laser-field induced currents in gases. a** Experimental setup (see text for details). OAP: off-axis parabola. A change of the wedge position $d_x$ results in a change of the amount of glass inserted in the beampath. The few-cycle pulses are shown for two CEP values that are shifted by $\pi$ (indicated by the thick and thin red lines). **b** Current dispersion scans obtained by moving the fused silica wedges when recording currents from electrode A (blue line) and B (red line). The line thickness corresponds to the standard deviation of three consecutive measurements. The inset shows a zoomed view (black bar in **b**) of the signal from electrode A with individual datapoints marked as dots and errorbars corresponding to the standard deviation.

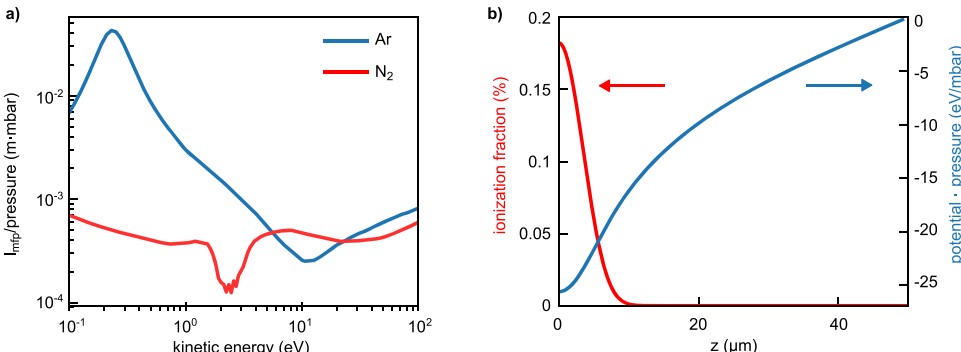

**Fig. 3 Theoretical modeling. a** Electron mean-free path $l_{mfp}$ at 1 mbar for argon (blue line) and nitrogen (red lines) as a function of kinetic energy. **b** Evolution of the ionization fraction (red line) and electrostatic potential (blue line) of the ion background toward the electrode for 1 mbar argon. Simulation parameters: beam waist $\omega_0 = 25\,\mu m$ and intensity $I = 1.2 \cdot 10^{14}\,W\,cm^{-2}$.

electrodes into a gas, where they reach intensities of $10^{14}\,W\,cm^{-2}$, and create a partially ionized ensemble of atoms and molecules. The creation and propagation of related charged particles induces currents in the electrodes. The setup is shown in Fig. 2a. Fused silica wedges are used to control the dispersion. The electrodes, denoted A and B in Fig. 2a, are produced from a copper wire with a diameter of ~500 μm. They are mounted individually on computerized stages, permitting distance control. The electrode assembly is placed on a linear closed-loop 3D-stage for fine positioning with respect to the ionization region. The positioning is monitored by an in-situ imaging system installed downstream. The focusing mirror and electrodes are located in a vacuum chamber, which allows us to vary the gas species (air, nitrogen, and argon) and control the pressure (0.1–1000 mbar) in the ionization region. The currents measured between each of the electrodes and ground are amplified by transimpedance amplifiers and detected via a two-channel lock-in amplifier. An example signal trace, obtained by scanning the dispersion with the wedges, is depicted in Fig. 2b for both electrodes. The observed oscillations are caused by the change of the CEP with the dispersion, while the envelope reflects the change in pulse duration and peak intensity. Since the electrodes A and B measure the current in opposite directions, their signals are $\pi$ out of phase.

**Theoretical model**. Our theoretical model is explained in detail in the Methods section. Briefly, for the numerical simulations of our experiments, an electrostatic PIC code was developed, which after ionization by the laser, considers the electron propagation under the influence of scattering. The laser is modeled with a Gaussian envelope in space and a pulse duration of 4.5 fs full-width-at-half-maximum (FWHM) of a Gaussian intensity envelope. In our Monte–Carlo approach, the initial positions of charges are randomly sampled from the spatio-temporal-resolved tunneling rate.

For each time step in the propagation, the electron-neutral (atom or molecule) scattering probability is calculated via the mean-free path $l_{mfp}$ and Monte–Carlo sampling. The mean-free paths for argon (blue line) and nitrogen (red line) used in the simulations are shown in Fig. 3a at 1 mbar. Electron-electron and electron-ion scattering are neglected due to the small ionization levels (<1%) in our experiments.

The charge interaction is calculated by projecting the electrons and ions on a grid, solving the Poisson equation and determining the resulting electrostatic field on each electron. Figure 3b shows the ionization fraction (red line) and the electrostatic potential of the ion background for 1 mbar argon (blue line) from the center between the electrodes toward one electrode. The strength of the ion potential also explains why related experiments that

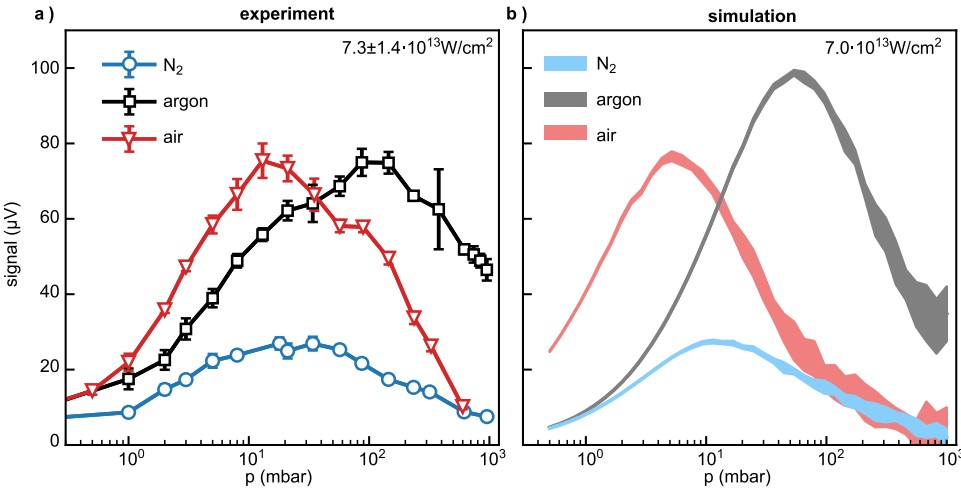

**Fig. 4 Pressure-dependence of the maximum current signal amplitude for nitrogen (blue), argon (black) and air (red). a** Experimental data for different gases. The symbols correspond to the datapoints and the errorbars correspond to the standard deviation of three consecutive measurements. The lines serve as a guide to the eye. **b** Simulation results. The width of the curves corresponds to the standard deviation of ten simulations with different random sampling of the initial electron distribution ($\omega_0 = 8\,\mu m$).

measured the generated charge via bias voltages, were conducted at low pressures[21] or had to apply kV-level biases.

Having established a simulation model that is able to self-consistently describe the motion of the charged particles after laser excitation, a method to determine the signal on the electrodes from the charge motion is required. Our rigorous approach to calculate how a moving charged particle induces a charge on an electrode is based on the Ramo–Shockley theorem[24,25], see Methods for details.

**Comparison of experimental and theoretical results**. The recorded pressure dependencies of the maximum current signal for nitrogen (blue line), argon (black line), and air (red line) are shown in Fig. 4a. The experimental data has been averaged over three individual dispersion scans (see Fig. 2b), and the errorbars correspond to the standard deviation. Performing the measurements via dispersion scans is necessary, since increasing the pressure leads to a shift of the maximum to lower glass insertions. The electrode distance was around $100\,\mu m$. A rather low intensity of $7.3 \cdot 10^{13}\,W\,cm^{-2}$ was chosen in order to avoid reshaping of the dispersion trace with increasing pressure. Starting from low pressure, all three curves are increasing and reach a maximum at different pressures, nitrogen at around $30\,mbar$, argon at $100\,mbar$, and air at $10\,mbar$. They subsequently decay going toward $1000\,mbar$. The maximum signal amplitudes in argon and air are roughly equal, whereas the amplitude is around a factor of three lower for nitrogen.

The simulations well reproduce the main features of the measured pressure dependence (cf. Fig. 4), especially the relative maximal amplitudes and pressures at which the maximum is reached. An intensity of $7 \cdot 10^{13}\,W\,cm^{-2}$, close to the experimental conditions, provides the best results from the simulations. Since the individual pseudo-electrons have a larger weight for higher pressure, a higher standard deviation of the Monte–Carlo simulations is obtained for higher pressures. The simulated distributions are slightly narrower than the experimental ones, which is likely due to the 2D-approximation.

Figure 5 shows the electrode-distance dependence of the maximum signal amplitudes in nitrogen for pressures of $10\,mbar$ (blue line), $100\,mbar$ (red line), and $530\,mbar$ (gray line). As above, each data point has been obtained from the average of three dispersion scans. The signal amplitude increases nonlinearly

by almost a factor of four when decreasing the distance from $420\,\mu m$ to roughly $30\,\mu m$. At even lower distances, the laser hits the electrodes, which we intentionally avoided. The simulations (light blue, light red and light gray areas, peak intensity $8 \cdot 10^{13}\,W\,cm^{-2}$) reproduce the distance-dependence above roughly $150\,\mu m$. They, however, slightly overestimate the signal for lower distances, which is further discussed below. For comparison, the $1/D$-dependence is also shown (dashed lines). It reproduces the behavior in both experiment and simulations for larger distances.

The scaling of the maximum signal strength with intensity can be seen in Fig. 6a for nitrogen (blue dots) and air (red triangles) at $25\,mbar$. The signal amplitudes grow rapidly by almost two orders of magnitude when doubling the experimental peak intensity from 4 to $8 \cdot 10^{13}\,W\,cm^{-2}$. For even higher intensities, the signal amplitude saturates manifesting as a kink in the signal vs. intensity graph. In air, saturation is reached at slightly lower intensities (which we attribute to the higher ionization from oxygen). Below saturation, the signal amplitude in air is about a factor of 3–5 higher than in pure nitrogen. The simulations (performed for $\omega_0 = 25\,\mu m$) for nitrogen (black crosses) and air (gray crosses) reproduce relative amplitudes and the initial transition from the rapid increase to saturation very well. A small systematic offset of around 20 % between the experimental intensity calibration (lower axis) and the intensity in the simulation (upper axis) is observed. For the lowest intensities, the simulations underestimate the measured signal. This can be traced back to the ionization model (see Methods) not being appropriate anymore in this regime[26].

To better illustrate the connection between signal saturation and the formation of the kink, the experimentally measured dispersion traces in nitrogen are shown in Fig. 6b (blue curves, left side) and compared to the simulated traces (black curves, right side). The temporal evolution of the laser pulse used in the calculation is obtained from a d-scan measurement. Again, overall good agreement is observed. The low-intensity wings of the traces are underestimated in the simulation which can largely be explained by the findings above. Most importantly, the saturation of the signal trace is reproduced. It is connected to amplitude quenching at high intensities (Fig. 6a). The simulations show that the occurrence of the kink is not related to a saturation of the ionization (or the vanishing CEP-effect[27]) but is a consequence of the collective effect of electron-neutral scattering

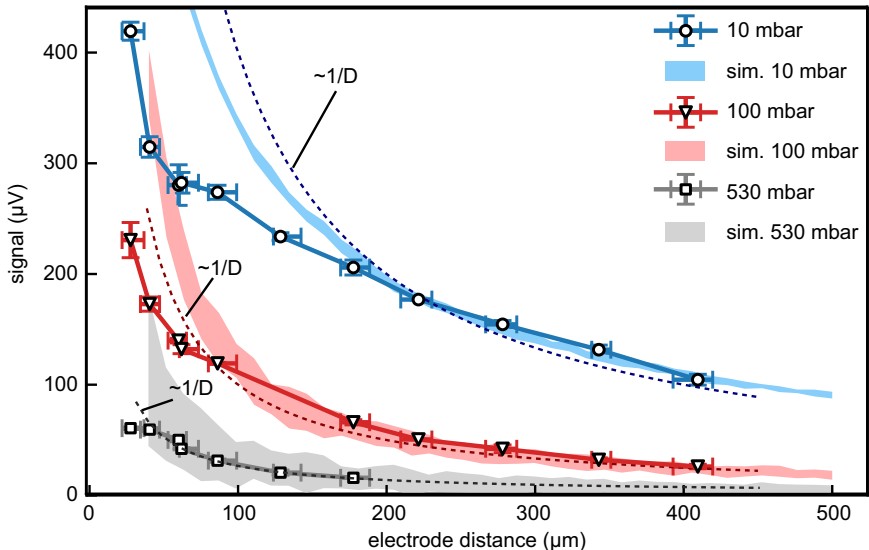

**Fig. 5 Electrode-distance dependence.** Maximum signal amplitudes for pressures of 10 mbar (blue), 100 mbar (red), and 530 mbar (gray) in nitrogen at $8.3 \pm 1.8 \cdot 10^{13}\ \mathrm{W\,cm^{-2}}$. The symbols correspond to the datapoints, the lines serve as a guide to the eye. The errorbars in $y$-direction correspond to the standard of three consecutive measurements. The errorbars in $x$-direction represent the minimum and maximum retrieved distance from the imaging of the electrodes for each distance. Simulation for 10 mbar (light blue area), 100 mbar (light red area) and 530 mbar (light gray area) for $\omega_0 = 8\ \mu m$, $I = 8.0 \cdot 10^{13}\ \mathrm{W\,cm^{-2}}$. The width of the curves represents the standard deviation of 18 (10 mbar), 24 (100 mbar) and 36 (530 mbar) simulations. For comparison the $1/D$-dependence is shown (dashed lines).

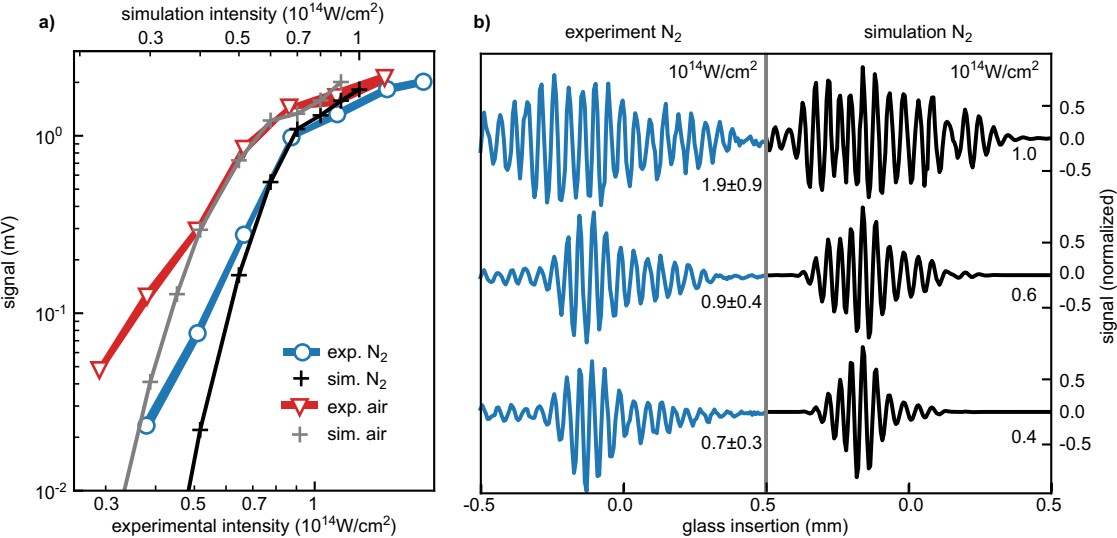

**Fig. 6 Intensity-dependence and signal trace reshaping. a** Intensity-dependence of the maximum signal amplitudes for nitrogen (blue dot) and air (red triangles) at a pressure of 25 mbar together with the simulations for nitrogen (black crosses) and air (gray crosses). In the experiments, an additional telescope (see Methods) and an electrode distance of around $100\ \mu m$ was used. The width of the experimental curves represents the standard deviation of three consecutive measurements. **b** Comparison of experimental (blue lines, left) and simulated (black lines, right) signal traces for different input intensities. The simulations used $\omega_0 = 25\ \mu m$.

and the rapidly increasing mean-field charge interaction which is further discussed in the following.

## Discussion
In order to illustrate the effects of electron-neutral scattering and the mean-field charge interaction, we investigate their influence on the signal formation. Figure 7a shows the simulated time-dependence of the induced charge on one electrode normalized by the total emitted charge for nitrogen. Here, scattering and charge interaction are selectively disabled by switching off the corresponding terms in the numerical propagation. With neither

scattering nor charge interaction (black line), the relative induced charge increases rapidly and reaches around 10 % as would be predicted from the photocurrent. Indeed, the initial slope of all three curves is proportional to the standard expression for the photocurrent $I = \sum q \cdot v$, where the right-hand side represents the sum over all charges $q$ and their velocity $v$ in detection direction. However, when scattering is enabled (blue line), the rise of the induced charge is quickly damped and reaches close to the asymptotic value of about 1 % after 0.2 ns. Qualitatively, this observation can be understood by considering that electron propagation leads to charge induction only up to the first (iso-tropic) scattering event. Afterwards, when neglecting that the

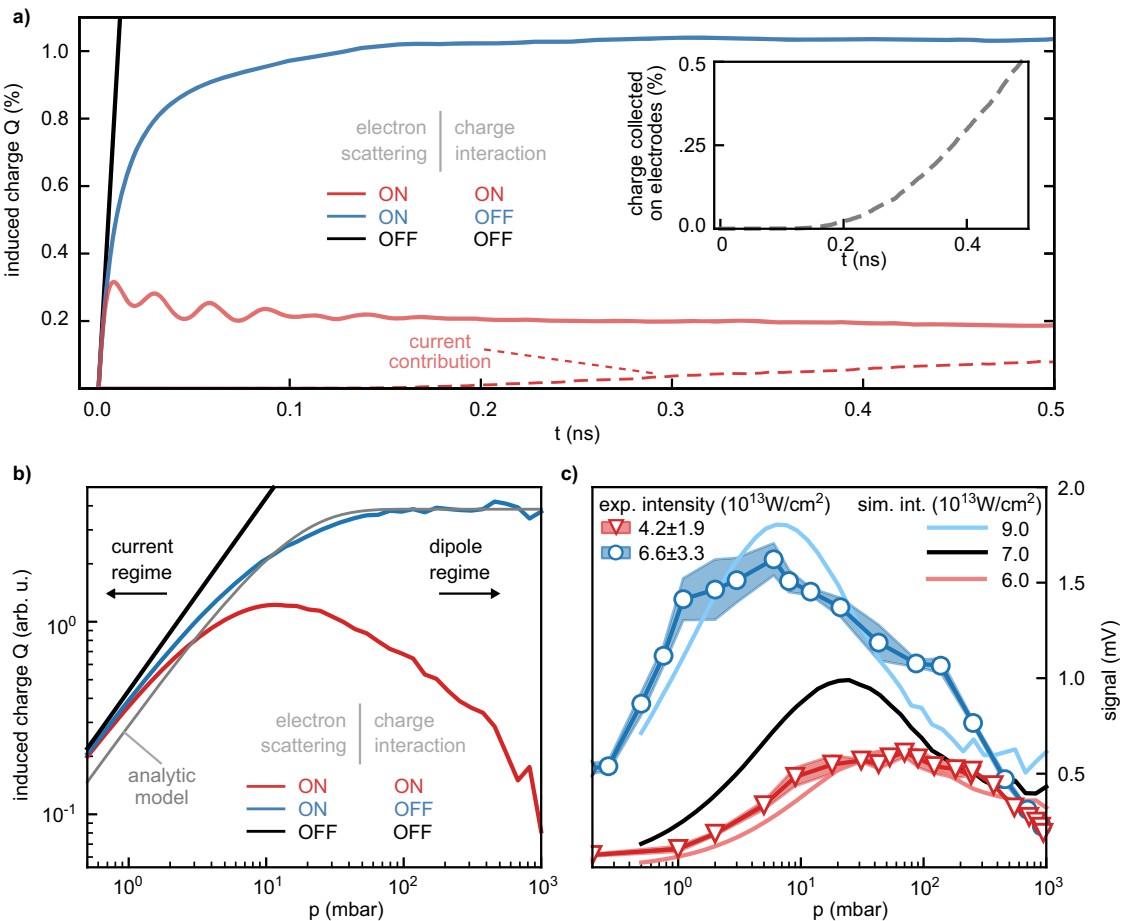

**Fig. 7 Role of electron-neutral scattering and the mean-field charge interaction. a** Time-resolved induced charge $Q$ at electrode A with selectively disabled scattering and charge interaction for 100 mbar nitrogen. The full simulation (red line) is compared to the case when only electron scattering (blue line) or neither scattering nor charge interaction (black line) are taken into account. The dashed red line shows the current contribution to the induced charge $Q$ for the full simulation, where the inset indicates the amount of free charges reaching electrode A. **b** Pressure dependence of the induced charge for nitrogen with selectively disabled scattering and charge interaction. The same color coding as in (**a**) is used. The gray line correspond to the analytic approximation in Eq. (1). **c** Experimental scaling of the pressure dependence with different laser intensities in argon (shaded areas) and comparison with simulations (solid lines). The width of the experimental curves represents the standard deviation of three consecutive measurements. Simulation parameters: (**a**, **b**) $\omega_0 = 8\,\mu m$ and $I = 0.7 \cdot 10^{14}\,W\,cm^{-2}$, (**c**) $\omega_0 = 25\,\mu m$.

electrodes may hinder further propagation, on average no charge is induced. If the charge interaction is switched on (red line), the rise is damped even faster and an asymptotic value of 0.2 % is reached. Here, the mean-field charge interaction counteracts the creation of a charge imbalance by generating a restoring force between the electrons and the ion distribution impeding the expansion of the electron ensemble. This restoring force manifests also in the initial, small and fast-decaying oscillations that can be seen on the charge signal which are caused by plasma oscillations (see Supplementary Note 1 and Supplementary Fig. 1). Figure 7a also illustrates why the experimental signal is calculated from the total induced charge at 0.5 ns after the initial rise when it reaches its asymptotic value. The transient initial current burst cannot be resolved experimentally.

In addition, we simulated the pressure dependence with and without both effects, cf. Fig. 7b. Without any interactions (black curve), the signal is simply proportional to pressure. Once scattering is considered, the signal saturates above around 50 mbar. Again, intuitively, for high pressures the contribution of a single charge is proportional to the distance it travels until the first scattering occurs, i.e., the mean-free path is scaling with $1/p$, since scattering is isotropic. On the other hand, the number of charges scales linearly with $p$, therefore, the signal is constant at high

pressures. Since the distance between the electron and the parent ion is limited by the electrodes, the signal drops once the mean-free path is on the order of the distance to the electrode. An analytic description of the strength of the induced charge $Q$, considering the electron distribution between the electrodes and the number of charges reaching the electrodes up to the first scattering event, is given by

$$Q \propto \frac{qp \cdot l_{\mathrm{mfp}}}{D}\left(1 - e^{-0.5D/l_{\mathrm{mfp}}}\right), \qquad (1)$$

which is shown in Fig. 7b (gray line). The best fit to the simulated curve is obtained by using a mean-free path approximately twice as large as the one used in the simulation. The above expression also suggests that instead of the velocity $v_x$ as for the photocurrent used by most models so far, the weighting factor for the macroscopic contribution of an individual electron should be the effective mean-free path $l_{\mathrm{eff}}$ in the electrode direction:

$$l_{\mathrm{eff}} = \frac{v_x \cdot l_{\mathrm{mfp}}}{\sqrt{v_x^2 + v_\perp^2}}, \qquad (2)$$

where $v_x$ and $v_\perp$ are the velocities in detection direction and perpendicular to it, respectively, and where $l_{\mathrm{mfp}}$ is energy-

dependent. These formulas should also be applicable to femto-second streaking[13] or PHz-scale nonlinear photoconductive sampling[14].

Accounting for the charge interaction makes electrons experience the attraction by the positive ion background. Intuitively, in the simplified picture above, after the first scattering event, the electron motion is not isotropic anymore but the emerging electrostatic field acts to undo the created charge imbalance. As a result, the average electron displacement $\langle x \rangle$ falls below the mean-free path. For a given intensity, this effect becomes more pronounced at higher pressures, since the concentration of free charges and the strength of the charge interaction grows proportionally with pressure. Consequently, the measured signal is maximal at the gas pressure that maximizes $p\langle x \rangle$, which for most intensities in our study is in the range of 1–10 mbar (see also Fig. 7c). An interesting question is how the maximum signal scales with the wavelength $\lambda$ of the driving laser. Since the average kinetic energy grows with $\lambda^2$ and the number of free charges with $\lambda$, the maximum signal increases with driving wavelength (see Supplementary Note 2, Supplementary Figs. 2 and 3).

We note that at the lowest pressures the signal becomes independent of whether electron scattering or charge interactions are considered, as all three curves in Fig. 7b converge. We identify this as the photocurrent regime, where practically all charges are able to reach the electrode. In contrast, at higher pressures, in what corresponds to the dipole regime, charge interactions can strongly affect the signal strength.

The influence of the charge interaction is further illustrated in Fig. 7c which shows the pressure dependence in argon measured for two peak intensities of $4.2 \cdot 10^{13}\,\mathrm{W\,cm^{-2}}$ (red triangles) and $6.6 \cdot 10^{13}\,\mathrm{W\,cm^{-2}}$ (blue dots), together with the corresponding simulations at $6 \cdot 10^{13}\,\mathrm{W\,cm^{-2}}$ (light red line) and $9 \cdot 10^{13}\,\mathrm{W\,cm^{-2}}$ (light blue line) and additionally at $7 \cdot 10^{13}\,\mathrm{W\,cm^{-2}}$ (black line). Again, a slight systematic shift between the experimental intensity compared to the simulation is observed. In both experimental and simulation data at higher intensities, the maximum signal grows, due to the increased number of charges. The mean-field charge interaction also grows, leading to a shift of the maximum to lower pressures. A similar effect is observed in Fig. 4, when comparing the pressure dependence of nitrogen and air at equal intensity. While very similar total scattering cross-sections can be assumed for both gases, the air data peaks at lower pressures since a higher number of charges is generated from oxygen. At the same time, once the peak intensity is so high that the maximum occurs at a pressure lower than in the experiment, saturation occurs. This, in turn, leads to a convolution of the intensity and pressure dependence.

Regarding the signal generation mechanism, there has been some debate on the roles of the asymmetric charge distribution (dipole contribution) compared to the charges that reach the electrodes (photocurrent contribution)[13]. In the Ramo–Shockley framework applied here, there is no real difference if a single charge is considered, as its relative contribution to the charge on the electrode smoothly reaches a value of one at the electrode surface (see Methods). This implies that the total charge $q$ of the particle is induced in the electrodes regardless of whether charge $q$ has entered the electrode or sits close to the surface. Moreover, for the idealized situation of infinite parallel plate electrodes, the induced charge $Q$ just scales linearly with decreasing distance of the particle to the electrode. In order to further clarify the roles of photocurrent and dipole contribution, we specifically looked at the amount of free charges $q_{\mathrm{current}}$ that reach the electrodes, as shown in the inset of Fig. 7a and their contribution to the induced charge (red dashed line, main panel). As can be seen, $q_{\mathrm{current}}$ constitutes a considerable fraction of the total free charge. Moreover, both the dipole contribution as well

as the photocurrent contribution take part in forming the total induced charge $Q$ that is measured in the end.

Despite its assumptions, our model provides surprisingly good agreement with the measurements. The model, however, has some limitations. The most important is the mean-field treatment of electrostatic interactions, that neglects electron-electron and electron-ion scattering as well as electron-ion recombination. The former two effects likely have a similar influence as electron-neutral scattering. For our experimental parameters with ionization degrees below roughly 1 %, this simplification seems justified. At higher intensities, a microscopic extension might be necessary, where additional reshaping of the laser pulse due to the generated plasma[28] is considered. Moreover, in our model the effect of ion movement has been neglected as well as the role of retardation effects, which are important for THz generation in plasmas[16–18]. Finally, at some point plasma produced light could lead to electron emission from the electrodes, which we neglected. Although not relevant for the current study, our approach can be extended toward higher intensities by using particle-particle-particle-mesh ($P^3M$) PIC-codes, which should be able to describe most of these effects.

In summary, we have investigated the emergence of macroscopic currents in photoconductive sampling of optical fields. Experimentally, we found that for most of our experimental conditions the pressure-dependent current signals for nitrogen, argon, and air show a maximum between 10 mbar and 100 mbar. Our quantitative model provides an accurate description of the measurements, uncovering the roles of the electron-atom/electron-molecule scattering and charge interaction. We found that the observed maximum in the current signal can be explained by a surprisingly large influence of the mean-field charge interaction. Our results show that while the heuristic photocurrent and dipole model can be thought of as limiting cases for low and high pressures, respectively, the signal generation at high pressures in the dipole regime is strongly influenced by the interplay of electron-neutral scattering and mean-field charge interaction. These findings present a way to boost the sensitivity of current measurements in gases, performed so far only at atmospheric pressures, by more than an order of magnitude. Our theoretical framework on macroscopic signal formation can be straightforwardly extended to other experimental scenarios and other media, including photoconductive sampling of electric fields in gases and solids.

## Methods

**Experimental**. We used a commercial Ti:Sa laser system (Femtopower HR/CEP-4) which provides up to 0.7 mJ pulse energy at 780 nm with 27 fs pulse duration at 10 kHz repetition rate. For spectral broadening the pulses are sent through a hollow-core fiber filled with argon at 0.5 bar. They are subsequently compressed using chirped mirrors to durations reaching 4.5 fs in FHWM of the intensity envelope at 750 nm central wavelength. The temporal intensity envelope was determined via the Dispersion Scan (d-scan) technique. Pulse energies up to only 18 $\mu$J have been used in the experiments.

The laser pulses of 4.5 fs duration at 750 nm are focused by an off-axis parabola (OAP, $f = 101.6$ mm) to a beam waist $w_0$ of below 10 $\mu$m. A mirror-based telescope can be introduced in front of the setup to increase the beam waist by about a factor of three. For the calibration of the intensity, we measured the focal spot size inside the experimental chamber via a charge-coupled device camera. Moreover, the relative peak intensity compared to the Fourier limit for our 4.5 fs laser pulses was determined from a d-scan measurement in front of the chamber. We obtained a conversion factor from the pulse energy, measured by a power meter, to the peak intensity in the experimental focus of $1.1 \cdot 10^{14}\,\frac{\mathrm{W}}{\mu\mathrm{J\cdot cm^2}} \pm 20\%$. For the situation including the telescope, a factor of $0.11 \cdot 10^{14}\,\frac{\mathrm{W}}{\mu\mathrm{J\cdot cm^2}} \pm 50\%$ is determined. Here, a higher relative uncertainty is obtained, since the telescope introduces a slight astigmatism, affecting the accuracy of the focal spot size determination.

The currents measured between each of the electrodes and ground are amplified by two transimpedance amplifiers (Femto DLPCA-200) with a gain of $10^9$ V A$^{-1}$. The resulting voltage pulses are detected via a two-channel lock-in amplifier (Zürich Instruments HF2LI).

In order to measure the CEP-dependence of the currents, the CEP is flipped between $\varphi_0$ and $\varphi_0 + \pi$ for consecutive laser pulses (indicated by the thick and thin red lines in Fig. 2a) using an acousto-optic dispersive programmable filter (Fastlite Dazzler). Consequently, the demodulation of the voltage signals in the lock-in amplifier is performed at half the repetition rate $f_{rep}/2$.

**Theoretical.** The signal on the conducting electrodes is formed by a simple electrostatic mechanism: The induced charge $Q$ on an electrode is given by the surface charge that is induced by the presence of a charged particle $q$. The change of the induced charge $Q$ can be measured as a current if the electrodes are connected to ground. In the simplest case of an infinite conducting plate, $Q$ is given by the value of the image charge.

Immediately after ionization, the electron and parent ion are still at the same position. Since their charges have opposite signs, the induced surface charges cancel. A net charge is induced only if one charge gets displaced with respect to the other. For practical applications, it would be cumbersome to calculate the induced surface charge for each position of the electron/ion and then integrate over the electrode surface. This approach would be feasible only for very simple, highly symmetric geometries. Fortunately, the calculation is considerably simplified through the Ramo–Shockley theorem[24,25].

Here, the induced charge $Q$ on the electrode and current $I$ flowing from the electrode, caused by a particle with charge $q$ at position $\vec{r}$ and velocity $\vec{v}$, are given by[22,23]:

$$Q = -q\phi_0(\vec{r}), \tag{3}$$

$$I = q\vec{v}\vec{E}_0(\vec{r}), \tag{4}$$

where $\phi_0$ is the weighting potential and $\vec{E}_0$ is the weighting field. For any arrangement of electrodes, the weighting potential can be calculated by setting the potential on the electrode under consideration to unity (1 V in SI units) and to zero on all other electrodes. For an ensemble of charges, the induced charge is given by the sum over the individual particle contributions.

For infinitely extended parallel plates, the weighting potential can be obtained analytically and has a very simple form: It linearly depends on the position of the charge between the electrodes. It is one at the electrode under consideration and linearly decays to zero at the other electrode. We use this idealized weighting potential (cf. Fig. 8a) in our simulations due to its simplicity. For comparison, we numerically calculated the weighting potentials for a realistic geometry of two opposing metallic cylinders with ratios of their distance $D$ to the radius $R$ of the cylindrical electrodes of $D/R = 2$ and $D/R = 6$, with the latter depicted in Fig. 8b. The electrodes were meshed using GMSH[29,30] and the electrostatic calculation was performed using the boundary-element implementation of scuff-em[31,32]. The calculated weighting potentials for the different $D/R$ ratios are shown in Fig. 8a along the cylinder axis for electrode A. Compared to the linear infinite plate solution (blue dashed line), the realistic weighting potentials (blue solid and dotted lines) decay faster when moving away from the electrode. The weighting potentials for electrode B (red lines) are symmetric around the center plane at $z = 0$.

Such realistic electrode configurations with more complex weighting potentials can be more sensitive to charges closer to the electrodes. In an intuitive picture, this is the case, if more electric field lines of the particle charge do not end up on the electrodes but escape to the surroundings. We note that this situation applies especially to solid-state experiments, where thin electrodes are deposited on the surface. The linearity of these measurements could be affected. The exact scaling of the signal with electrode distance[19,20] will in such cases depend on the actual electrode geometry. Additional effects might play a role in the macroscopic signal formation, such as the surface roughness of the electrodes or potentially dielectric

passivation layers which could modify the weighting potentials. In principle, information on the weighting potentials could directly be obtained in future, carefully designed experiments with well characterized electrodes and modeling of the whole electric circuit. Here, the laser focus between the electrodes could be scanned while measuring both the current signal from the individual electrodes after amplification as well as the lockin-demodulated CEP-dependence. Such a characterization would have general importance to assess signal formation in ultrafast current measurements. For the experiments discussed here and their numerical modeling, however, we find that as long as the electrode distance remains limited compared to the surface dimension of the electrodes, and the laser focus remains centered between the two electrodes, the infinite plates approximation is well justified.

Due to computational limitations, the simulations had to be restricted to two spatial dimensions perpendicular to the laser beam propagation direction. We find this to be a good approximation because the focal beam waist $\omega_0$ is much smaller than the Rayleigh length $z_R$. The electrons are modeled as an ensemble of pseudo-particles with an effective charge given by the total emitted charge divided by the number of pseudo-particles $N$. For the results presented here, we use $N = 5 \cdot 10^5$. The total charge is obtained by radially integrating the final ionization fraction calculated by the Ammosov-Delone-Krainov (ADK) rate[33] multiplied by the atomic number density ($\propto$ pressure $p$). For nitrogen ($I_p = 15.58$ eV) the same tunneling rate as for argon ($I_p = 15.76$ eV) is employed. In order to model the contribution of oxygen in air, which has a much lower ionization potential ($I_p = 12.56$ eV)[34] than nitrogen, we use the ADK-parameters of xenon ($I_p = 12.13$ eV), but with angular momentum quantum numbers of $l = 2$ and $m = 1$. The latter is important since it takes into account the symmetry of the molecular wavefunction of $O_2$ in the tunneling region, which leads to a significantly lower tunneling rate than in xenon ($l = 1$, $m = 0$)[34].

In our Monte–Carlo approach, the initial position of a pseudo-electron and the corresponding pseudo-ion is randomly sampled from the spatio-temporal-resolved tunneling rate. From the latter, the final emission velocity of the pseudo-electron is calculated in the SMM within the SFA under the assumption that effectively only direct electrons contribute. Pseudo-electrons have a charge-to-mass ratio of $e/m_e$ such that they behave like normal electrons during propagation. Pseudo-ions are assumed to stay fixed at the birth position.

The propagation of pseudo-electrons is performed via the Velocity-Verlet algorithm[35] using a time-step of 20 fs, or smaller if required, over a time-span of 1 ns. For each time step, the electron-neutral (atom or molecule) scattering probability is calculated via the mean-free path $l_{mfp}$. $l_{mfp}$ is obtained from the MagBoltz[36] cross-sections available via the xcat-database[37], that contain elastic, excitation and ionization cross-sections. The mean-free paths for argon (red line) and nitrogen (blue line) at 1 mbar are shown in Fig. 3b. The mean-free-path in both argon and nitrogen are similar above 10 eV; however, it increases in argon below the threshold of ionization and excitation. In nitrogen, on the other hand, excitation channels at lower energies due to the molecular structure result in a minimum of $l_{mfp}$ at around 2 eV. Differences between the two gases reach more than an order of magnitude at electron kinetic energies around 0.5 eV. It is important to note that the mean-free path scales with $1/p$, so $l_{mfp}$ is on the order of 1 mm for 1 mbar, while for 1000 mbar it is around $1\,\mu m$. The scattering time is above $10^{-13}$ s even at atmospheric pressure. For the inelastic channels we assume a uniform probability for the energy loss from the threshold of the inelastic process, e.g., the ionization potential for the ionization channel, up to the electron kinetic energy. Secondary electrons are neglected. For simplicity, we assume isotropic scattering in the lab frame for all processes. We assume the atoms to be fixed, i.e., a temperature of $T = 0$ K for the gas, which is justified since the laser-driven electron kinetic energy (around 1 eV) and the corresponding velocity is significantly higher than the kinetic energy of thermal motion of the atoms (around 40 meV) and the

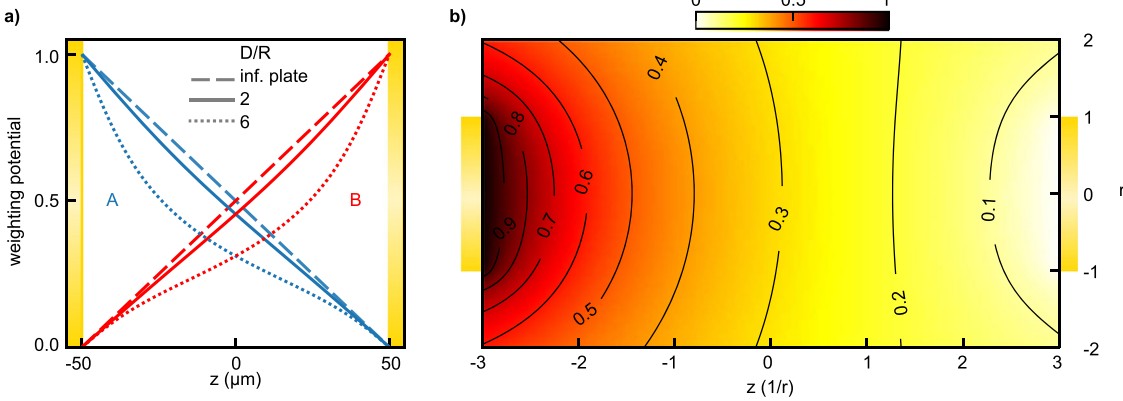

**Fig. 8 Weighting potentials. a** Ramo–Shockley weighting potentials for an electrode distance $D = 100\,\mu$m and different ratios of the distance $D$ to the electrode radius $R$ for cylindrical electrodes A (blue) and B (red). **b** Radially-resolved weighting potential for the left electrode for $D/R = 6$.

atomic velocity. Moreover, we only simulate a limited time interval during which no significant energy transfer between electrons and ions due to scattering occurs. Since we deal with ionization fractions of around 1% and below, scattering and recombination with the ions is neglected. For air, the scattering cross-sections of nitrogen are used.

In order to calculate the electrostatic mean-field interaction, the Poisson equation is solved on a 2D-grid for each time step. The length of the rectangular simulation region in the $x$-direction is the electrode distance $D$. Along the $y$-direction, the length is taken to be $3D$. The grid contains 512 points along the electrode axis and 1536 along the other axis. The laser focus is positioned in the center of the rectangle. The pseudo-electrons are sampled on the grid using a linear weighting scheme. We impose the Dirichlet boundary condition for the potential $\phi = 0$ at all four edges of the simulation region using the method of image charges. Toward this end, the grid is doubled in size and a charge of opposite sign is injected at the position mirrored along the positive edge. Due to the implicit periodicity of the Fast-Fourier Transform used for solving the Poisson equation, all contributions of the otherwise infinite sum of mirror charges are contained in the calculation. The electric field is obtained in the same step and linearly interpolated onto the positions of the pseudo-electrons. If a particle leaves the simulation region it is not considered anymore in the electrostatic interaction. Across the boundary perpendicular to the electrodes the propagation is continued whereas it is stopped if the pseudo-electron reaches the electrodes. For the pseudo-ions, the potential and field calculation on the grid is only calculated once at the start of the simulation.

At each time step, the induced charge $Q$ on both electrodes is calculated using the linear weighting potential of the infinite parallel plates shown above and summing over the ensemble of pseudo-electrons. Since the induced current decays over a timescale of 100 fs–1 ps (cf. discussion in relation to Fig. 7), much faster than the bandwidth of measurement electronics, the measured signal is assumed to be proportional to the induced charge $Q$ averaged from 0.5–1 ns after the start of the simulation, when $Q$ has reached a quasi asymptotic value (see Fig. 7a). The lower bound does, however, not impact the results significantly. In order to obtain the signal in the experiment from the 2D-simulation, the simulated induced charge density is multiplied by the repetition rate of the laser (10 kHz), the transimpedance gain ($10^9$ V/A) and the effective ionization length $\Delta z_{ion,eff}$ which is a free parameter, whose order of magnitude should be between the focal spot-size and the Rayleigh length. The used values are $\Delta z_{ion,eff} = 7\,\mu m$ (Fig. 4b), $\Delta z_{ion,eff} = 44\,\mu m$ (Fig. 5), $\Delta z_{ion,eff} = 47\,\mu m$ (Fig. 6) and $\Delta z_{ion,eff} = 25\,\mu m$ (Fig. 7c).

## Data availability
The data that support the findings of this study are stored on a server of the Max Planck Society under restricted access. The access is restricted to limit the use of the data by a data use agreement to reproduction of the results of this study only. Inquiries for data access should be sent to the corresponding author and will typically be responded to within a matter of days.

## Code availability
The code used for the simulations contained in this study is available from the corresponding author upon reasonable request. The access to the proprietary code is restricted through a code use agreement to reproduction of the results of this study only. Related requests will typically be responded to within a matter of days.

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

## Acknowledgements
We thank Nicholas Karpowicz for fruitful discussions, and are grateful for support by Ferenc Krausz. The work was carried out at LMU/MPQ and we acknowledge related support by the German Research Foundation (DFG) via SFB NOA (Z.W., B.B. and M.F.K.) and LMUexcellent (M.F.K.), by the European Union via FETopen PetaCOM (A.M., Z.W., V.Y., B.B., and M.F.K.) and FETlaunchpad FIELDTECH (A.M., J.B., Z.W., B.B., and M.F.K.), and by the Max Planck Society via the IMPRS for Advanced Photon Science (J.S., A.M., and D.Z.) and via the Max-Planck Fellow program (M.F.K.). We acknowledge support by the King-Saud University in the framework of the MPQ-KSU collaboration (J.S., Z.W., M.A., A.M.A., and M.F.K.). We are grateful for support by the Researchers Supporting Project number (RSP-2021/152), King-Saud University, Riyadh,

Saudi Arabia (M.A., and A.M.A). M.F.K.'s work at SLAC is supported by the U.S. Department of Energy, Office of Science, Basic Energy Sciences, Scientific User Facilities Division, under Contract No. DE-AC02-76SF00515.

## Author contributions

M.W., V.Y., B.B., and M.F.K. directed the project. J.S., A.M., J.B., and D.Z. carried out the experiments and analyzed the data. J.S. developed the theoretical model and performed all simulations. P.R. helped with operating the laser. Z.W. contributed to the implementation of the experimental setup and the discussion. M.A. and A.M.A. contributed to the discussion. J.S. wrote the paper, which was finalized with input from all authors.

## Funding

## Competing interests

The authors declare no competing interests.
