## [Peer Review File · Nature Communications]

The emergence of macroscopic currents in photoconductive sampling of optical fieldsREVIEWER COMMENTS

Reviewer #2 (Remarks to the Author):

Schotz and coworkers present an in-depth study on the understanding of currents measured at electrodes placed close to a laser gas interaction region. Here, laser-induced ionization takes place, resulting in currents in the electrodes. Recently, such currents have been shown to contain details of the laser field. Most importantly, the carrier envelope phase of individual laser pulses can be measured this way, in rather simple setups. So far, the exact origin of the measured current was elusive, which is the main driving force for this study. Hence, this work is timely.

The results are based on a combination of pressure-dependent current measurements and comparison to PIC simulations. In addition, the authors invoke the Shockley Ramo theorem for a more insightful understanding. The results are a deeper understanding on how the measured electrode current relates to the ionized gas (and its density plus environmental conditions).

The paper contains nice and clear figures, but I am sorry to say that it is not written too well (see a few details below). Chiefly, the manuscript could have been more concise and is not too easy to follow.

Summing up, I am also sorry to say that I cannot recommend publication in a journal addressing a broader science audience. This nice work is clearly of technical relevance for other groups working in the field and should hence be published, but in a more technical journal.

The authors might want to take these recommendations into account:

- Frequently, the authors mention “roles of electron scattering and Coulomb interactions.” Clearly, electron scattering is based on Coulomb interaction between two electrons. In the course of the manuscript it becomes clear that “Coulomb interactions” seems to refer to the interaction of the photo-emitted electrons with their parent ions. This should be all written up in a much clearer fashion.
- P. 3, bottom: “current I on the electrode” – this doesn’t seem meaningful.
- P. 6, third line of Discussion: “selectively disabled” – this is an important point that certainly deserves more discussion (technically how, which terms?).
- P. 6, last line: “pulls the electron back” – how is this inferred?
- P. 7 top: “connected to plasma oscillations”: does the frequency match the expected plasma frequency?
- Fig. 7 a, inset: it is not clear if the % current is % of the induced charge (so % of %) or “units” like Fig. 7a.
- P. 8, center: “there is no real difference on the single charge level” – this deserves a more in-depth and physics discussion. Currently, it is not clear what exactly the authors mean.

- P. 8, bottom: "as shown in Refs. 22, 23": I couldn't find that these works discuss electrode shapes. Please check or re-phrase.

- P. 9, center: The authors mention that "heuristic models utilized so far need to be amended". It is not clear by what. This should be worked out in greater clarity.

Nature Communciations web site is extremely slow, which makes it painful to submit a report.

Reviewer #3 (Remarks to the Author):

1. If the wavelength of the incident laser is changed, will the signal collected by the two electrodes change? If so, what is the law of change?
2. Will the generated ions react with the copper electrodes to affect the experimental results when you did the experiments.
3. The temperature of measurement and simulation in this manuscript should be given, because the current signal may vary with temperature. It would be better if the simulation model can take temperature into account.
4. Why the electrodes used in the simulation model are infinitely extended parallel plates, but cylindrical electrodes have been used in the actual measurement?
5. The simulation result of 530 mbar should be added to in Figure 5 like the 10 mbar and 100 mbar.
6. Please give the reason why the deviation between simulation and experimental results in Figure 5 is large when distance is small. Moreover, please explain the phenomenon that the deviation of simulation and experimental results under 10 mbar condition is obviously larger than that under 100 mbar condition.

Response to the reviewers:

We thank the reviewers for their comments, which have prompted us to obtain further data and helped to significantly improve the presentation of our work. We address the comments point-by-point below. The original comments are in **black**, while our responses are highlighted in **blue**. Since the revisions to the text are very substantial, we highlight some of them in **green**.

Reviewer 2:

Schotz and coworkers present an in-depth study on the understanding of currents measured at electrodes placed close to a laser gas interaction region. Here, laser-induced ionization takes place, resulting in currents in the electrodes. Recently, such currents have been shown to contain details of the laser field. Most importantly, the carrier envelope phase of individual laser pulses can be measured this way, in rather simple setups. So far, the exact origin of the measured current was elusive, which is the main driving force for this study. Hence, this work is timely.

The results are based on a combination of pressure-dependent current measurements and comparison to PIC simulations. In addition, the authors invoke the Shockley Ramo theorem for a more insightful understanding. The results are a deeper understanding on how the measured electrode current relates to the ionized gas (and its density plus environmental conditions).

We appreciate the evaluation of the reviewer that our study is timely and provides a deeper understanding of how macroscopically measured currents are related to the generated charges and their interactions in optoelectronic field sampling.

The paper contains nice and clear figures, but I am sorry to say that it is not written too well (see a few details below). Chiefly, the manuscript could have been more concise and is not too easy to follow.

We thank the reviewer for the positive feedback on the figures and critical feedback on the text. We have carefully addressed this in our revision, where we very significantly revised the text and moved technical details of both experiment and theory into the Methods section. We are convinced that the revised version is now much more concise and easier to follow also for a non-specialist.

Summing up, I am also sorry to say that I cannot recommend publication in a journal addressing a broader science audience. This nice work is clearly of technical relevance for other groups working in the field and should hence be published, but in a more technical journal.

We apologize that the previous writing style of the manuscript might have helped to leave this impression. We have addressed this in the revision. We also humbly disagree with this evaluation of the interest in our work. Optoelectronic field sampling is a technique with wide-spread applications and our work has already generated very high interest from others, who are not limited to detecting currents in gases but for example also in solids. Indeed, the general concept and conclusions reached in our study are highly relevant also in other scenarios. We are therefore very strongly convinced that this work, which contains a full theoretical framework carefully tested against experimental results that enables an understanding of current formation in realistic optoelectronic field sampling devices, will attract a wide audience.

The authors might want to take these recommendations into account:

- Frequently, the authors mention “roles of electron scattering and Coulomb interactions.” Clearly, electron scattering is based on Coulomb interaction between two electrons. In the course of the manuscript it becomes clear that “Coulomb interactions” seems to refer to the interaction of the

photo-emitted electrons with their parent ions. This should be all written up in a much clearer fashion.

We thank the referee for pointing out that possible source of confusion. With “electron scattering”, we were referring to the scattering of electrons and atoms (scattering on ions can largely be ignored for our ionization levels), while by “Coulomb interaction”, we meant mean-field charge interaction amongst the electrons and between the electrons and ions.

We have changed all instances of Coulomb interaction into “mean-field charge interaction”, or brief “charge interaction” and also added electron-neutral scattering for most instances, this also includes the in-figure label of Fig. 7. Also, in the description of the simulation, we have tried to point this out more explicitly [see paragraphs starting with “For each time step in the propagation, the electron-neutral (atom or molecule) scattering probability...” and “The mean-field charge interaction is calculated...”].

- P. 3, bottom: “current I on the electrode” – this doesn’t seem meaningful.

We have moved this section to the Methods and changed the sentence to:

p.9 “Here, the induced charge Q on the electrode and current I flowing from the electrode...”

- P. 6, third line of Discussion: “selectively disabled” – this is an important point that certainly deserves more discussion (technically how, which terms?).

We appreciate the comment by the reviewer. How the “Coulomb interaction” and “electron scattering” are treated in the simulations is described in detail in the Methods section. We have expanded the description of the charge interaction and scattering (see also response above) and we have altered the sentence as follows:

p.6 “Here, scattering and charge interaction are selectively disabled by switching off the corresponding terms in the numerical propagation”

- P. 6, last line: “pulls the electron back” – how is this inferred?

We appreciate the comment by the reviewer. This can be inferred in several ways from the simulations. First of all, the observed plasma oscillation (see also answer below) indicates that the electrostatic force between the ions and electrons creates a restoring force that equilibrates charge imbalances created by the movement of the electrons. Secondly, this can be even more directly inferred by studying the electron distribution and its temporal evolution with and without charge interaction (e.g. in the simulations for Fig. 7). In the former case, a large portion of the electron distribution stays localized around the ions.

We have slightly adapted the statement to make it more precise:

p.6: “Here, the mean-field charge interaction counteracts the creation of a charge imbalance by generating a restoring force between the electrons and the ion distribution impeding the expansion of the electron ensemble. This restoring force manifests also in the initial, small and fast-decaying oscillations that can be seen on the charge signal which are caused by plasma oscillations (see Supplementary Note 1 and Supplementary Fig. S1).”

In the framework presented here, there is no real difference if a single charge is considered, as the weighting potential (see Methods) of the electrode smoothly reaches a value of one at the electrode surface.

- P. 7 top: “connected to plasma oscillations”: does the frequency match the expected plasma frequency?

We thank the referee for the interesting question. We have added Supplementary Note 1 and Supplementary Fig. S1 which discuss the plasma oscillations in more detail. In brief, we have determined the center frequency of the initial oscillations by using a Fourier-transformation as illustrated in Fig. S1a) and b) for a wide range of different pressures and intensities. As can be seen in Fig. S1c), the extracted oscillation frequency is approximately equal to half the plasma frequency f_{pi} calculated from the peak charge density n_{max} . This finding backs our statement, that the fast-decaying oscillations that can be seen on the charge signal stem from the plasma oscillation. The reason that the observed oscillation frequency does not exactly match the plasma frequency of the peak charge density can be explained by the inhomogeneous initial charge distribution caused by the finite laser spot-size which leads to a rapid drop of the charge density away from the center. Moreover, electron scattering from atoms leads to energy and momentum relaxation and thereby affects the electron dynamics differently for changing pressures.

We have included a reference to the discussion of the plasma oscillation in the Supplementary Information in the revised manuscript:

p.6 “This restoring force manifests itself also in the initial, small and fast-decaying oscillations that can be seen on the charge signal which are caused by plasma oscillations (see Supplementary Note 1 and Supplementary Fig. S1).”

- Fig. 7 a, inset: it is not clear if the % current is % of the induced charge (so % of %) or “units” like Fig. 7a.

We appreciate the comment by the referee since this can indeed cause some confusion. The % current is the % of the total free charges that reach the electrodes. For improved clarity we have changed the label to “charge collected on electrodes (%)”. Note that the total induced charge is smaller than the charge that reaches the electrode, since the leftover ions and electrons propagating in the other direction lead to a contribution to the induced charge with an opposing sign. In conclusion, the induced charge is determined by the asymmetry of the evolving electron distribution.

- P. 8, center: “there is no real difference on the single charge level” – this deserves a more in-depth and physics discussion. Currently, it is not clear what exactly the authors mean.

We appreciate the comment by the reviewer. We believe that by connecting the statement with the two explanatory sentences that were following it, it becomes clearer. According to the weighting potential, there is no difference whether the electron sits just in front the electrode surface (still counting as dipole contribution) or enters the electrode (counting as current contribution), which means there is nothing special about an electron reaching the electrode. At the end it just matters that the charge distribution expands asymmetrically.

We have changed the sentences as follows:

p. 8: “In the Ramo-Shockley framework applied here, there is no real difference if a single charge is considered, as its relative contribution to the charge on the electrode smoothly reaches a value of one at the electrode surface (see Methods). This implies that the total charge q of the particle is induced in the electrodes regardless of whether the charge q has entered the electrode or sits close to the surface.”

- P. 8, bottom: "as shown in Refs. 22, 23": I couldn't find that these works discuss electrode shapes. Please check or re-phrase.

The references were referring to the first part of the sentence "...scaling of the signal with electrode distance". This part has been moved to the Methods and to make this immediately more clear, we have changed the sentence to:

p.9 "The exact scaling of the signal with electrode distance[19, 20] will in such cases depend on the actual electrode geometry."

- P. 9, center: The authors mention that "heuristic models utilized so far need to be amended". It is not clear by what. This should be worked out in greater clarity.

The thank the reviewer for this comment. We have slightly changed the wording at the corresponding positions to make it more in line with the results of our work:

In the abstract we have changed the wording to:

p.1 "The results show that the heuristic models utilized so far are valid only in a limited range and are affected by macroscopic effects."

We have changed the wording in the conclusion to:

p.8 "Our results show that while the heuristic photocurrent and dipole model can be thought of as limiting cases for low and high pressures, respectively, the signal generation at high pressures in the dipole regime is strongly influenced by the interplay of electron-neutral scattering and mean-field charge interaction."

Additionally, we have added a paragraph in the discussion comparing the pressure dependence with and without scattering and charge interaction:

p.8 "We note that at the lowest pressures the signal becomes independent of whether electron scattering or charge interactions are considered, as all three curves in Fig. 7b) converge. We identify this as the photocurrent regime, where practically all charges are able to reach the electrode. In contrast, at higher pressures, in what corresponds to the dipole regime, charge interactions can strongly affect the signal strength."

Reviewer 3:

1. If the wavelength of the incident laser is changed, will the signal collected by the two electrodes change? If so, what is the law of change?

We thank the referee for this interesting question. We have added Supplementary Note 2 and Supplementary Figs. S2 and S3, where we discuss this in more detail. In short, we have performed simulations for both argon and nitrogen, where we changed the central wavelength λ_0 from 300 nm to around 2000 nm. The pulse duration was set to 1.8-cycles and the peak intensity was kept fixed at $0.8 \cdot 10^{14}$ W/cm². With the employed tunnel emission model this leads to a linear increase with λ_0 of the number of free electrons (as long as ionization saturation can be neglected). At low pressures, in the current regime, where electron scattering and charge interaction can approximately be neglected, this leads to a linear increase of the signal. At the same time, the average kinetic energy of the electrons increases by λ_0^2 since it scales with the ponderomotive energy. This results in a change of the average electron mean-free-path. At higher pressures, the scaling of the signal strength is thus affected by the energy-dependence of the mean-free path (see Fig. 3a in the main text) which explains the difference between nitrogen and argon. While in nitrogen the mean-free path stays almost constant with increasing energy, it drastically reduces for argon. The latter leads to a relative reduction of the signal strength. At higher pressures the scaling of the signal strength is more complicated and is influenced by the mean-free path and charge interaction.

2. Will the generated ions react with the copper electrodes to affect the experimental results when you did the experiments.

We thank the reviewer for the comment. The detailed role of the ions in the experiment is still to be clarified. In our simulations, we assumed the ions to act as a static, positive charge background with which our experimental results are very well reproduced. This simplifying assumption is mainly based on the different velocities of electrons and ions as well as the limited time window (1 ns) considered in our simulations.

While, in principle also the ions gain momentum from the laser pulse (in the opposite directions of the electrons), the kinetic energy from the light interaction is on the order of 20 μ eV, whereas the thermal energy is around 40 meV for $T = 300$ K. Therefore, in contrast to the electron ensemble, the initial velocities of the ions are not determined by the asymmetry of the laser field but rather the isotropic thermal movement, which does not lead to a CEP-dependent current signal.

The average velocity of the ions, dominated by thermal movement is on the order of 100 m/s. In order to induce a strong signal on the electrodes, the ions have to experience a displacement on the length scale of the electrode separation. Even in the ballistic regime at low pressures, this would mean that the timescale of an induced ion signal is on the order of 100 ns to 1 μ s, which is beyond the timescale of our simulations. Moreover, at these timescales, additional processes like electron-ion recombination will play an important role (see e.g. A. E. Martirosyan et al, J. Appl. Phys. 96, 5450 2004).

The generated ions can in principle react with the copper electrodes surface through absorption. The electrodes are expected to naturally possess a passivation layer from the interaction with the background gas and also from their preparation under atmosphere. Indeed, atomically clean surfaces would require special in-situ cleaning procedures and vacuum levels below 10^{-10} mbar to maintain the cleanliness over extended periods of time which is not feasible for our experiments. However, we do not expect any surface layers to drastically affect our measurements. Since the ion energy is low, effects such as plasma etching that would alter the electrode shape, can likely be ignored.

In conclusion, we don't expect the ions to significantly contribute to the laser field-dependent current signal but to generating a charge balance on longer timescales that is independent of the laser field waveform. A detailed investigation of the role of the ions will be left to a future study.

3. The temperature of measurement and simulation in this manuscript should be given, because the current signal may vary with temperature. It would be better if the simulation model can take temperature into account.

We thank the reviewer for this comment as it raises an important point. In our simulation we assume a gas at zero temperature, i.e. the gas atoms and parent ions are at rest. The effect of a finite gas temperature would be two-fold in our simulations. First, it would change the velocity of the initial electrons. Secondly, in the scattering events with the gas atoms, the electrons would gain or lose momentum even for elastic scattering. To estimate whether a finite temperature-effect would play a role for our experiments, it is instructive to compare the electron velocity v_{el} to the velocity of the center of mass v_{cm} (electron + atom/molecule) in the lab frame.

$$\vec{v}_{cm} = \frac{m_{el}\vec{v}_{el} + m_{atom}\vec{v}_{atom}}{m_{el} + m_{atom}},$$

where v_{atom} and m_{atom} are the velocity and mass of the atom (we implicitly include molecules). Only if v_{cm} would have the same order of magnitude as v_{el} , a strong temperature influence would be expected:

$$|v_{el}| \leq |v_{cm}| \leq \left(\frac{m_{el}}{m_{el} + m_{atom}}\right) \cdot |v_{el}| + \left(\frac{m_{atom}}{m_{el} + m_{atom}}\right) \cdot |v_{atom}|,$$

where we have used the triangle inequality to obtain the right-hand side. Anticipating $m_{atom} \gg m_{el}$, the following simplified condition for strong temperature effects can be derived in terms of kinetic energies:

$$E_{kin,el} \leq \frac{m_{el}}{m_{atom}} E_{kin,atom}.$$

For room-temperature $T \sim 295K$, we obtain a kinetic energy for the thermal motion of the atoms of around 0.04 eV. Moreover, for both argon and nitrogen, we obtain $m_{el}/m_{atom} < 2^{-5}$. For the typical electron kinetic energies for our laser parameters on the order of 0.1 eV to 1 eV the equation above is not fulfilled. The temperature-dependence can thus be neglected. This justifies the simplifying assumption used in our simulations for the limited timescales considered there. Nevertheless, the gas temperature will likely be included in future versions of the code.

We have added a comment on the temperature used in the simulation in the Methods section:

p. 11: "We assume the atoms to be fixed, i.e. a temperature of $T=0$ K for the gas, which is justified since the laser-driven electron kinetic energy (around 1 eV) and the corresponding velocity is significantly higher than the kinetic energy of thermal motion of the atoms (around 40 meV) and the atomic velocity. Moreover, we only simulate a limited time interval during which no significant energy transfer between electrons and ions due to scattering occurs."

4. Why the electrodes used in the simulation model are infinitely extended parallel plates, but cylindrical electrodes have been used in the actual measurement?

The infinitely extended parallel plates in the simulation model lead to a very simple analytic expression for the weighting potential of each electrode. Using the realistic cylindrical electrode geometry requires the calculation of the weighting potentials using an electrostatic solver for each electrode distance. For electrode distances smaller than the electrode diameter, the weighting potential between the cylinders is still well described by the approximation of the infinitely extended parallel plates. To show this, we have included now also the weighting potential for $D/R=2$ in Figure 8a) in the Methods. Furthermore, to illustrate the agreement even further, we have extended our simulations where we have included the realistic Ramo-Shockley weighting potential for a typical electrode distance of $100\ \mu\text{m}$ (electrode radius $250\ \mu\text{m}$). Here, the weighting potentials were calculated with the approach described in the main text, on a rectangular grid in z and r direction between the cylinders. For each electron at each time step the weighting potential has been obtained by linear interpolation from the grid points. As illustrated in Fig. R1 below, for different pressures excellent agreement of the induced signal between the full Ramo-Shockley model (orange dashed line) and the approximation (black line) is observed. In conclusion, theoretically, as long as the electrode distance is limited compared to the surface dimension and the laser focus is centered between the two electrodes, the infinite plates approximation is justified.

Experimentally, however, very large parallel plates (independent of e.g. whether they are square-shaped or cylindrical) lead to a high capacitance at the entrance of the transimpedance amplifier which can strongly affect its performance.

Fig. R1: Comparison of the induced signal for two different Ramo-Shockley potentials. The electrostatic calculation for the more realistic geometry of two cylinders facing each other has been done for a cylinder radius $r=250\ \mu\text{m}$ and a distance $D=100\ \mu\text{m}$. Excellent agreement between the infinite plate approximation (black line) and the full electrostatic simulation (orange dashed line) is observed as illustrated for different pressures of 1.6 mbar (a), 11.3 mbar (b) and 53.8 mbar (c). Other simulation parameters are $I=0.8 \cdot 10^{14}\ \text{W}/\text{cm}^2$, $\omega_0=8\ \mu\text{m}$.

5. The simulation result of 530 mbar should be added to in Figure 5 like the 10 mbar and 100 mbar.

We thank the reviewer for the suggestions. We have added the 530 mbar simulation result to the figure and included it in the caption.

6. Please give the reason why the deviation between simulation and experimental results in Figure 5 is large when distance is small. Moreover, please explain the phenomenon that the deviation of

simulation and experimental results under 10 mbar condition is obviously larger than that under 100 mbar condition.

We appreciate the comment by the reviewer. There are several possible effects that could play a role. First of all, our simulations are performed in 2d rather than 3d. Secondly, the electrodes in the experiment show imperfections compared to the infinite plate but also the opposing cylinder model (discussed above in the context of the realistic Ramo-Shockley potential). For instance, the electrodes are neither perfectly parallel nor completely smooth. For lower pressures and lower distances, when the electrons get displaced further and easier reach (close to) the electrodes, the signal could potentially be more affected. Finally, we so far did not consider the buildup of an electrostatic field between the two electrode surfaces as the electrodes charge up capacitively. This might create an additional force that counteracts the electron motion. Again, this effect would be the strongest for lower pressures and lower distances. To add this effect into the simulation, the macroscopic electrical circuit would need to be modeled, which we leave to future studies.

REVIEWERS' COMMENTS

Reviewer #2 (Remarks to the Author):

The authors have done a lot of work improving the manuscript, which is why I recommend to publish.

Reviewer #3 (Remarks to the Author):

This work can be accepted without further revision needed